# The Interleukin-8-CXCR1/2 Axis as a Therapeutic Target in Peritoneal Carcinomatosis

**DOI:** 10.3390/curroncol32090496

**Published:** 2025-09-05

**Authors:** Christopher Sherry, Neda Dadgar, Zuqiang Liu, Yong Fan, Kunhong Xiao, Ali H. Zaidi, Vera S. Donnenberg, Albert D. Donnenberg, David L. Bartlett, Patrick L. Wagner

**Affiliations:** 1Allegheny Health Network Cancer Institute, Pittsburgh, PA 15224, USA; christopher.sherry2@ahn.org (C.S.); neda.dadgar@ahn.org (N.D.); zuqiang.liu@ahn.org (Z.L.); yong.fan@ahn.org (Y.F.); kunhongkevin.xiao@ahn.org (K.X.); ali.zaidi@ahn.org (A.H.Z.); albert.donnenberg@ahn.org (A.D.D.); david.bartlett@ahn.org (D.L.B.); 2Department of Cardiothoracic Surgery Pittsburgh, University of Pittsburgh School of Medicine, Pittsburgh, PA 15260, USA; donnenbergvs@upmc.edu; 3Department of Medicine, Drexel University School of Medicine, Philadelphia, PA 19104, USA

**Keywords:** peritoneal carcinomatosis, immune environment, interleukin-8 (IL-8)/CXCL8, CXCR1/CXCR2, maladaptive, neutrophil, angiogenesis, epithelial–mesenchymal transition (EMT), targeted therapy, biomarker

## Abstract

Peritoneal carcinomatosis (PC) is a serious condition where cancer spreads throughout the lining of the abdominal cavity. It often occurs in late stages of cancers like ovarian, colon, or stomach cancer and is hard to treat. Patients with PC usually suffer from complications such as fluid buildup (ascites), bowel blockages, and poor nutrition, which severely impact quality of life and limit treatment options. This review focuses on a molecule called interleukin-8 (IL-8), which is part of the body’s immune system but also helps cancers grow and spread. IL-8 attracts immune cells like neutrophils that, instead of fighting the cancer, create an environment through its receptor CXCR1/CXCR2 signaling that helps tumors survive, invade new areas, and avoid immune attack. IL-8 also encourages inflammation, blood vessel growth to feed tumors, and scar tissue buildup that causes bowel obstruction. Blocking IL-8—either with specific antibodies or drugs that stop its action—has shown promise in early lab studies and clinical trials. These treatments may help reduce cancer spread, ease symptoms like ascites, and improve how well other cancer therapies work. Our review concludes that targeting IL-8 and its receptor CXCR1/CXCR2 is a promising new strategy for treating peritoneal carcinomatosis and could improve survival and quality of life for patients. More research is needed to confirm its benefits and develop combination treatments.

## 1. Introduction

Peritoneal carcinomatosis (PC) is a late manifestation of abdominopelvic malignancies characterized by metastasis within the peritoneal cavity [1,2]. Although any tumor type can theoretically metastasize to the peritoneal cavity, the most common sites of tumor origin are the digestive or gynecologic tracts. Less common, but equally difficult to manage, are tumors metastatic from extra-abdominal organs as well as rare primary neoplasms of the mesothelial lining itself (e.g., mesothelioma and primary peritoneal carcinomatosis). Considering all the various sites of tumor origin, it has been estimated that over 70,000 patients per year are affected by PC in the United States alone [3,4]. PC has a poor prognosis, and treatment strategies often rely on palliative chemotherapy and surgery. However, recent investigations are focusing on the peritoneal immune environment as a potential target for immunotherapy. Here, we review the role of interleukin-8 (IL-8) in PC progression and its potential as a target for PC treatment.

Recent studies have highlighted the unique biochemical environment of the peritoneal cavity in PC and demonstrated that this compartment *significantly influences* cancer progression and disease-related morbidity and mortality [5,6,7,8,9,10]. The peritoneal cavity is a sequestered immune environment, which differs in soluble and cellular immune elements relative to the systemic circulation [1,11]. Moreover, the peritoneal environment differs dramatically in carcinomatosis relative to the physiological state [1]. The cytokine milieu in malignant peritoneal fluid is skewed toward a maladaptive polarity that favors non-specific innate immunity and inflammation over adaptive cytotoxic antitumor immunity [1,2,12]. Driven by a proinflammatory secretome, the PC environment facilitates epithelial–mesenchymal transition (EMT) of and immune evasion by metastatic cancer cells [1,12]. EMT promotes motility and enables tumor cells to migrate from the primary site throughout the peritoneal cavity [1,2,12], whereas mesenchymal-to-epithelial transition (MET) drives adhesion and epithelialization. Tumor cells are often in a partial EMT state (pEMT), that enables rapid adaptation and metastatic seeding [13]. This mechanism leads to colonization and invasion of mesothelial surfaces, a hallmark of PC. These mechanisms contribute to the pathobiology of PC and are an area of intense ongoing research given the high demand for innovative treatment options in these advanced cancer settings [1,3].

We and others have reported on the soluble and cellular components of the peritoneal immune environment [1,14,15], and found significantly elevated proinflammatory mediators such as interleukin-6 (IL-6), interleukin-8 (IL-8), fibroblast growth factor 2 (FGF2), and monocyte chemoattractant protein-1 (MCP-1 or CCL2) in the peritoneal fluid of PC patients [1]. Concomitantly, malignant peritoneal fluid is characterized by low concentrations of adaptive immune markers, such as interferon-γ (IFNγ), interferon-α (IFNα), interleukin-1 (IL-2), and interleukin-10 (IL-10), suggesting that the immune landscape in PC favors innate immunity and suppresses tumor-specific immune responses. This dysregulation of the immune response is complex and involves the interplay of multiple cytokines, including IL-8, to promote the development of cancer. The proinflammatory cytokine milieu, including IL-6 and IL-8, contributes to PC’s immune evasion mechanisms [1] and is under investigation as a therapeutic target [16].

The current standard of care for PC often involves systemic chemotherapy and cytoreductive surgery in fit patients with amenable tumor biology. Although aggressive therapy leads to prolonged survival, or even remission in some individuals, most patients eventually progress and develop late complications of PC, including bowel obstruction and malignant ascites. In addition to having few treatment options, patients at this late stage generally experience malnutrition and cachexia, which further limits their eligibility for clinical trial enrollment [5,6,16,17]. Since the peritoneal cavity is sequestered by peritoneal mesothelial lining from systemic circulation, regional immunotherapy has gained traction as a potential adjunct to systemic therapy [16,18]. However, regional immunotherapy modalities have not achieved clinical utility, to date, in spite of investigation into monoclonal antibodies, cytokines, adoptive cellular therapies, and other options as candidate therapies [1,9,19,20,21]. However, the most opportune targets and practical combination of therapies remain to be defined, warranting more in-depth translational investigations.

Since both IL-6 and IL-8 are potent mediators of inflammation, their elevated presence in the peritoneal cavity suggests that they play a role in the maladaptive, chronic inflammation seen in PC [22]. IL-8, an important proinflammatory cytokine and a key mediator of neutrophil recruitment and degranulation, has also been found to have direct and indirect implications in pro-tumorigenic pathways [23,24,25]. Given this cytokine’s inflammatory and tumorigenic potential, it is hypothesized that treating PC patients using an IL-8 blockade would be beneficial [1,26,27]. Ongoing studies are exploring the potential of IL-6 blockade as a treatment for combating PC’s inflammatory and oncogenic effects [16,17] and may hold implications for IL-8 inhibition or synergistic benefit in combination blockade. This review summarizes the role of IL-8 in PC progression and its potential as a therapeutic target.

## 2. Physiologic Functions of IL-8

IL-8, also known as CXCL8, is an essential chemokine that orchestrates inflammatory and immune responses, but its significance extends beyond innate immunity to pathological conditions such as chronic inflammation and fibrosis. Identified in 1987 as a neutrophil-activating cytokine, it has since been recognized for its broader roles in immune cell recruitment, activation [28] and tumor EMT [25]. IL-8 belongs to the CXC chemokine family and acts primarily through two G-protein-coupled receptors, chemokine receptor type 1 (CXCR1) and chemokine receptor type 2 (CXCR2), which are abundantly expressed on neutrophils [28,29].

A diverse range of cells produce IL-8, including immune cells such as monocytes, macrophages, neutrophils, and T lymphocytes, as well as non-immune cells like endothelial cells, fibroblasts, keratinocytes, epithelial cells, hepatocytes, and tumor cells. IL-8 production is triggered by various stimuli, including bacterial endotoxins (e.g., lipopolysaccharides), cytokines (e.g., IL-1 and TNF-α), and cellular stress signals [30,31]. The specific stimuli and cell types influence the intensity and duration of IL-8 secretion. For example, phagocytes and endothelial cells produce IL-8 robustly in response to endotoxins, whereas mesenchymal cells are relatively insensitive to endotoxin-driven IL-8 secretion [30]. Production by neutrophils is most robust in response to opsonized particles, allowing for juxtacrine secretion, additional neutrophil recruitment, and amplification of the inflammatory cascade [30]. The synthesis of IL-8 is tightly regulated by signaling pathways such as the interleukin-1 (IL-1)–interleukin-1 receptor (IL-1R)–TNF receptor-associated factor 6 (TRAF6) pathway and the tumor necrosis factor alpha (TNF-α)–tumor necrosis factor receptor (TNFR)–TNF receptor-associated factor 2 (TRAF2) pathway, both of which converge on nuclear factor kappa B (NF-κB) activation [24,32], allowing for IL-8 gene transcription, protein synthesis and the release of the cytokine into the extracellular space where it exerts its biological effects.

Once secreted, IL-8 binds to its receptors, CXCR1 and CXCR2. These receptors are expressed on various immune cells—including neutrophils, monocytes, T cells, and natural killer (NK) cells—as well as non-immune cells like endothelial and epithelial cells [29,30]. CXCR1 is present on both CD8+ cytotoxic T cells and CD4+FoxP3+ regulatory T cells, while CXCR1 and CXCR2 are found on CD14+ monocytes, CD56dimCD16+ NK cells, and basophils [33]. Additionally, CXCR1/2 expression on neurons and mesenchymal cells plays a role in angiogenesis and pain regulation [24]. IL-8 binding to its receptors initiates G-protein signaling, leading to G-protein subunit dissociation. The β and γ subunits of the G-protein activate effectors like PI3Kγ, PLCβ, and small GTPases (Ras, Rac, Rho), while the Gαi2 subunit inhibits adenyl cyclase, reducing cAMP levels [24,30]. These cascades elevate intracellular calcium, activate protein kinase C, and drive processes such as actin polymerization, cell adhesion and migration. In neutrophils, this signaling enhances chemotaxis, granule exocytosis, respiratory burst- through generation of reactive oxygen species (ROS), and neutrophil extracellular trap (NET) formation, supporting pathogen clearance and tissue repair [28,30,32,34]. IL-8 directs neutrophils to sites of injury or infection where they adhere to endothelial cells, transmigrate into tissues, and release proteolytic enzymes like elastase and myeloperoxidase to degrade pathogens and damaged tissues. IL-8 strengthens neutrophil attachment to the vascular endothelium and promotes migration to inflamed tissues by further upregulating adhesion molecules such as CD11b/CD18 [28,30,32,34].

While IL-8-driven inflammation is crucial for host defense, sustained production can lead to maladaptive responses and detrimental outcomes. Persistent IL-8 signaling can result in chronic inflammation, characterized by continuous neutrophil recruitment and activation, which damages host tissues. This pathobiology is observed in chronic conditions such as rheumatoid arthritis, inflammatory bowel disease, and chronic obstructive pulmonary disease [29,31,35]. Chronic inflammation often progresses to fibrosis, marked by excessive deposition of collagen and extracellular matrix components that disrupt normal tissue architecture, as seen in pulmonary fibrosis, liver cirrhosis, and systemic sclerosis. IL-8 also directly activate fibroblasts, contributing to tissue remodeling by direct activation of M2-like tissue macrophages whose activity releases matrix metalloproteinases (MMP) involved in extracellular matrix degradation, chronic tissue remodeling and wound healing. Chronic inflammation within the peritoneal cavity drives maladaptive fibrosis development, adhesive scar development, and subsequent bowel obstructions. Further, in inflammatory conditions like peritonitis and endometriosis, IL-8 orchestrates key pathological mechanisms [36,37,38,39]. In endometriosis, IL-8-driven angiogenesis promotes the proliferation and maintenance of ectopic endometrial tissue, perpetuating inflammation and contributing to symptoms such as pelvic pain and infertility [29]. In severe cases, such as sepsis or extensive trauma, IL-8 contributes to systemic inflammatory response syndrome, characterized by a cytokine storm, capillary leakage, and multi-organ failure [29]. In addition to these pathological effects, chronic IL-8 signaling can drive metabolic dysregulation, deplete nutritional reserves, impair tissue repair, and accentuate the development of cachexia [40,41,42].

In Summary, IL-8 often acting in concert with signaling molecules like TNFα and transcription factors like NF-kβ, is a double-edged sword in inflammation, immunity, tissue remodeling, pain, and wound healing. While essential for neutrophil-mediated defense against infections, dysregulated of chronic IL-8 activity can lead to significant pathological consequences, including chronic inflammation, fibrosis, and systematic immune dysregulation. A deeper understanding of the molecular mechanisms governing IL-8 activity offers opportunities for targeted therapies to mitigate its harmful effects in inflammatory and neoplastic diseases.

## 3. IL-8 and Cancer Progression

IL-8, often in concert with other cytokines like TNFα, significantly influences various aspects of cancer biology across multiple cancer types, including melanoma, prostate, colon, breast, lung, and pancreatic cancers [43]. It exerts both direct oncogenic influence on tumor cells, as well as indirect effects on the tumor microenvironment (TME). Together, these effects promote increased proliferation, resistance to apoptosis, enhanced cell motility, and angiogenesis. IL-8 alters the local immune response, recruits neutrophils, and polarizes the environment towards innate immunity while suppressing adaptive, tumor-specific cytotoxic responses, resulting in a microenvironment that supports tumor survival and progression [1,23,44]. Reviews of these mechanisms are found in Table 1 and Figure 1.

IL-8 promotes tumor cell proliferation and resistance to apoptosis by activating signaling pathways such as protein kinase B (AKT pathway). Pasquier et al. demonstrated that IL-8 plays a pivotal role in creating a pro-metastatic niche following surgical peritoneal stress, promoting cancer cell survival and resistance to apoptosis [45]. IL-8 was found to be responsible for resistance to apoptosis through the activation of the AKT pathway and chemoresistance in the observed ovarian cancer cells, which persisted through the establishment of an autocrine IL-8 loop [45]. IL-8 activates upstream pathways culminating in NFκ-β signaling, a driver of tumor proliferation, inflammation, and signaling [46,47,48,49]. In colorectal cancer (CRC), estrogen-related receptor α (ERRα) has been shown to be overexpressed and increases the expression of IL-8 and in turn facilitates tumor cell proliferation and migration. ERRα inhibition by XCT-790 significantly reduced IL-8 levels and suppressed tumor growth, demonstrating the cytokine’s central role in CRC biology [50].

Through regulating cellular adhesion, actin polymerization, and cytoskeletal dynamics, IL-8 enhances tumor cell motility and invasion. This is mediated by downstream activation of small GTPases, which regulate cell adhesion, actin polymerization, membrane protrusion, and eventually cell migration [23]. IL-8 also activates tumor associated macrophages (TAMs), whose activity releases MMP and contributes to invasive track generation, tumor EMT and invasion. This allows for cancer cell motility that may disseminate and metastasize. IL-8 also drives cytoskeletal rearrangements that enhance chemotaxis, a critical step for invasion. Neutrophils and TAMs, recruited by IL-8, secrete enzymes and proteases that degrade tissue barriers, facilitating tumor invasion of mesothelial surfaces [51].

IL-8 also plays a central role in angiogenesis, a key process in tumor growth, by directly stimulating endothelial cells to form new blood vessels that supply tumors with oxygen and nutrients [23]. Neutrophils recruited by IL-8 enhance this effect by releasing proangiogenic factors such as vascular endothelial growth factor (VEGF) and Bv8, while tumor and TAM-secreted matrix metalloproteinase-9 (MMP-9) degrades the extracellular matrix (ECM), releasing sequestered VEGF to further promote vascularization [50,52,53]. In colorectal cancer models, IL-8-driven angiogenesis increases vascular permeability and disrupts endothelial junctions, facilitating tumor extravasation [50,52]. Granulocyte-macrophage colony-stimulating factor (GM-CSF) from tumor cells recruits myeloid and polymorphonuclear cells and induces neutrophils to release oncostatin M (OSM), activating the JAK-STAT pathway and further amplifying VEGF expression and effects [53]. FGF2, stored in neutrophils and the ECM, is also released via neutrophil-secreted heparanase, contributing further angiogenic support of tumor and metastasis [53].

IL-8 has also been shown to be pivotal in immune modulation within the TME, promoting immune evasion by recruiting and activating neutrophils, myeloid-derived suppressor cells (MDSCs), TAMs, and cancer associated fibroblasts (CAFs) [23,44]. Through PI3K/AKT signaling, IL-8 promotes MDSC-mediated arginase I (ARG1) expression, inhibiting CD8+ T-cell cytotoxicity, while TAMs secrete factors that drive immune invasion [25] and immune escape [23]. IL-8 also stimulates fibroblasts, modifying extracellular matrix collagen deposition and creating a protective tumor-supportive stroma [23]. Neutrophils recruited to the TME differentiate into tumor-associated neutrophils with immunosuppressive and tumor-promoting functions, releasing ROS, NET, and cytokines that suppress tumor-infiltrating lymphocytes and facilitate metastasis [53]. Similarly to MDSC, neutrophils produce ARG1, depleting L-arginine and impairing T-cell function, while prostaglandin E2 and lipid peroxidation further suppress antigen presentation by dendritic cells [53]. Tumor-derived factors like TNF-α, GM-CSF, and IL-6 enhance PD-L1 expression on neutrophils via JAK-STAT3 signaling, directly inhibiting T-cell-mediated antitumor responses [53]. In addition to immunosuppression, neutrophils contribute to chronic inflammation and genetic instability by releasing ROS and proinflammatory microRNA that damage DNA and enhance cancer cell proliferation. They also support metastasis by interacting with circulating tumor cells, facilitating adhesion to distant organs, and supplying metabolic resources [53]. In gastric cancer, tumor-derived IL-8 promotes lymph node metastasis by upregulating PD-1 expression on CD8+ T cells, weakening their cytotoxic function [44].

## 4. IL-8 as a Cancer Biomarker

Given the wide cellular range of production and activity of IL-8, the local and systemic concentration of this molecule has been evaluated as a potential biomarker for use in clinical oncology. Elevated IL-8 levels are rarely observed in the serum of healthy individuals [30] but are markedly increased in cancer patients, including PC patients, and correlate with advanced disease and reduced survival [55,56,57,58]. While the evidence strongly supports IL-8 as a biomarker in digestive and gynecologic malignancies, its role remains primarily correlative, and mechanistic insights behind this association are limited. We speculate that the prognostic value of IL-8 is likely a surrogate for the increased tumor-supporting environment or systemic inflammatory state created by IL-8 secretion. As circulating levels of IL-8 increase, it is likely that regional or local secretome levels of IL-8 increase in the TME as well [59], culminating in pro-tumorigenic effects by driving cellular proliferation, resistance to apoptosis, and tumor dissemination [59]. Elevated IL-8 levels also represent the systemic inflammatory state characteristic of patients with advanced malignancies, contributing to increased catabolic state, nutritional decompensation, and overall demise [41,42], which is more pronounced in PC. Further research is needed to interrogate whether IL-8 is a causal contributor to disease progression, or merely a correlate of advanced malignancy.

Given the strong connection between IL-8 and neutrophils, it is perhaps unsurprising that the peripheral blood neutrophil-to-lymphocyte ratio (NLR) has emerged as a significant prognostic biomarker in cancer. Elevated NLR reflects systemic inflammation, a hallmark of cancer progression and poor prognosis. In metastatic CRC, a high NLR (>5) is independently associated with poor prognosis, regardless of other prognostic factors and molecular alterations. This elevation correlates with increased levels of inflammatory and angiogenic cytokines, underscoring NLR’s role in cancer progression [60]. Pre-treatment NLR serves as a valuable prognostic tool, with higher levels predicting worse survival outcomes; however, the optimal NLR cut-off may vary based on cancer type and timing of assessment [61,62,63]. In gastrointestinal cancers, including those with carcinomatosis, meta-analyses have consistently linked elevated NLR to worse overall survival (OS), disease-free survival, and progression-free survival (PFS) across different tumor sites and stages [64]. NLR is often considered superior compared to other inflammatory markers such as the platelet-to-lymphocyte ratio and lymphocyte-to-monocyte ratio in terms of predicting cancer outcomes. In colorectal cancer, NLR demonstrates higher predictive reliability, better concordance, and a more substantial predictive capacity for survival and cancer-specific survival [64]. NLR’s predictive value is further enhanced when combined with other biomarkers. For example, integrating NLR with carcinoembryonic antigen (CEA) and C-reactive protein improved prognostic accuracy in CRC [65]. As further investigations shed light on the predictive and therapeutic outcome value of IL-8 levels as a biomarker, the stand-alone utility and in combination with other biomarkers, like NLR, will hopefully become standard in treatment algorithms to gauge prognosis and steer treatment options, especially for patients suffering from PC.

Beyond prognostication, IL-8 has shown to be a predictive marker for therapeutic outcomes in non-PC patients that may inform PC treatment. In multiple studies [66,67,68] of breast cancer, lower baseline serum IL-8 levels were associated with improved survival and reduced chemoresistance [69]. Similar findings were reported in patients with hepatocellular carcinoma, where baseline IL-8 levels have been identified as predictive biomarkers for treatment response and survival [26]. In the SORAMIC trial, Öcal et al. demonstrated that baseline IL-6 and IL-8 levels predict the response to sorafenib as well as predict overall survival, supporting the role of IL-8 as a potentially predictive biomarker for optimizing treatment in hepatocellular carcinoma [70]. Elevated serum IL-8 levels have also been associated with reduced efficacy of immunotherapy in cancer patients [71,72]. Furthermore, IL-8 contributes to an immunosuppressive TME, aiding immune evasion and therapy resistance [23,44,53]. Future research is much needed to determine the causal mechanisms of IL-8 in cancer pathobiology and treatment.

In summary, serum IL-8 and NLR are robust, cost-effective biomarkers that provide valuable prognostic insights across multiple cancer types and should be used to monitor and inform therapeutic decisions in PC. Further research is essential to elucidate the mechanistic role of IL-8 in oncogenesis and its predictive relevance in guiding therapy.

## 5. IL-8 as a Driver of Peritoneal Carcinomatosis

PC represents an advanced stage of malignancy characterized by widespread tumor dissemination within the peritoneal cavity. IL-8 facilitates PC progression by promoting EMT in tumor and mesothelial cells by modulating the TME [73]. Chronic IL-8 stimulation sustains a cycle of inflammation and tumor proliferation, exacerbating disease progression and complications such as malignant bowel obstruction and ascites that are both common in patients with PC (Figure 2). Here, we depict the mechanisms by which IL-8 contributes to PC and evaluate preclinical trials that targeted IL-8.

Several potential mechanisms account for the ability of IL-8 to drive EMT in PC [54]. IL-8 has been shown to promote tumor progression in several cancer subtypes, including ovarian, breast, non-small cell lung, esophageal, and mesothelial cancer [25,73,74]. Other neutrophil-derived factors, such as IL-17, transforming growth factor-β (TGF-β), and neutrophil elastase (NE), further amplify EMT and tumor progression [53]. In gastric cancer, exosomal high-mobility group box 1 (HMGB1) triggers neutrophil autophagy via the HMGB1/TLR4/NF-κB pathway, releasing pro-tumor factors like IL-1β and OSM that enhance EMT and migration [53]. Even after primary tumor removal, EMT and inflammation continue to contribute to cancer recurrence. Persistent inflammation likely transforms dormant occult cancer cells into invasive tumors through NE and MMP-9 release during NET formation. These enzymes degrade adhesion proteins, release ECM sequestered growth factors, activating integrins like α3β1 to drive recurrence [53]. Once carcinoma has disseminated throughout the peritoneal cavity, IL-8 binds to CXCR1 on tumor cells triggering the signaling cascade and amplifying aggressive tumor growth, metastasis, and fibrosis [46,47,48,75,76], ultimately leading to tumor growth, disease progression, peritoneal spread in PC.

IL-8 is also critical to modulating an immunosuppressive environment within the peritoneal cavity. By recruiting neutrophils and MDSC, IL-8 enhances immune evasion in PC by directly inhibiting cytotoxic T-cell responses and supporting maladaptive would healing and inflammation. As IL-8 levels are elevated in the secretome of peritoneal fluid in the setting of PC [1], activated neutrophils are continuously stimulated within the peritoneal cavity to release various proinflammatory cytokines, proteases, and ROS, remodeling the extracellular matrix, promoting angiogenesis, and amplifying a pro-tumorigenic microenvironment that supports tumor growth and further dissemination in PC [77]. The release of NETs traps freely disseminated tumor cells in peritoneal fluid, facilitating their adhesion and invasion into the peritoneal cavity [78,79]. Neutrophils sequestered at tumor foci also modulate the integrity of the endothelial barrier, making it more permeable and permissive to extravasation of tumor cells, facilitating tumor cell dissemination [51].

IL-8 also contributes to systemic complications of PC such as ascites, fibrosis, and malignant bowel obstructions. The accumulation of protein-rich ascitic fluid, due to increasing vascular permeability and upregulation of pro-angiogenic factors like VEGF, Bv8, and OSM [50,52,53] leads to patient discomfort and impaired organ function, particularly the heart and kidneys, due to significant volume shifts [80,81]. Persistent IL-8 signaling also promotes peritoneal fibrosis, a key factor in developing malignant bowel obstruction—a leading cause of morbidity in PC [82,83,84,85]. Chronic inflammation activates fibroblasts, producing excessive extracellular matrix deposition, peritoneal thickening, and structural rigidity [23]. Because of this maladaptive environment, surgical intervention often exacerbates adhesions and can lead to bowel resections, further compromising intestinal function. IL-8-induced capillary leakage also potentiates chemotherapy-related side effects [86], further challenging the management of affected patients. Frequent hospitalizations and invasive procedures to address malignant ascites add complexity to the disease management, but seldom provide durable relief of pain and discomfort. Targeting the IL-8 axis could mitigate these complications by reducing neovascularization, decreasing fluid leakage, and alleviating symptoms associated with ascites and obstruction.

## 6. Preclinical Evidence Targeting the IL-8 Pathway

Due to its multifaceted role in PC pathogenesis, IL-8 has become an appealing target for therapeutic intervention in oncology (Figure 2). Inhibiting IL-8 or its receptors could disrupt inflammatory feedback loops sustaining the TME, reduce neutrophil and MDSC recruitment, and restore adaptive immune responses. IL-8 has been compared to other notable chemotactic agents, such as C5a, fMet-Leu-Phe, platelet-activating factor, and leukotriene B4 [30]. Unlike these agents, which are rapidly inactivated by oxidation or hydrolysis, IL-8 exhibits sustained activity [30,87]. IL-8 blockade could not only limit tumor proliferation and dissemination, but may alleviate systemic complications such as malignant ascites, fibrosis, and catabolic wasting. These changes would improve patients’ quality of life, alleviate debilitating symptoms, and increase eligibility for investigational therapies. The ultimate result would be better clinical outcomes for patients with PC.

A key potential drawback to IL-8-directed monotherapy is the redundancy of cytokines produced simultaneously in pathological conditions, suggesting that other cytokines with overlapping functions that will remain unaffected by IL-8 blockade. For instance, the elevations of both IL-8 and IL-6 found in PC have been the subject of current clinical investigation [16,17,73,88]. In the study by Alraouji et al., the authors investigated the effects of tocilizumab, an IL-6 receptor alpha agonistic antibody, on IL-8 production and the proangiogenic potential of triple-negative breast cancer cells [89]. The methods included treating triple-negative breast cancer cell lines with tocilizumab and measuring IL-8 through ELISA and RT-qPCR. Additionally, the study also assessed the impact on angiogenesis by conducting tube formation assays using human umbilical vein endothelial cells (HUVEC). The results showed that tocilizumab significantly reduced IL-8 production in triple-negative breast cancer cells, and treated HUVEC cells exhibited a marked decrease in angiogenesis, as evidenced by the reduced tube formation in vitro. By reducing IL-8 and angiogenesis, as well as reversing tumor EMT [22], tocilizumab contributes to a less aggressive tumor phenotype, highlighting its promise as an adjunctive treatment in managing malignancies.

IL-8 blockade has been studied in animal models [90,91]. In a dextran sodium sulfate-induced model of acute colitis in mice, those expressing the human IL-8 gene developed more tumors that were associated with higher quantities of myeloid cells that promote angiogenesis and tumor growth [29]. This finding reinforces the effects of IL-8, potentially impacting neutrophils and endothelial cells to promote tumorigenesis. In another study, Li et al. investigated an anti-IL-8 antibody in a humanized pancreatic cancer mouse model [44], finding enhanced myeloid cell activation and potentiation of the antitumor activity of an anti-PD-1 antibody. Such findings highlight the potential synergy between IL-8 blockade and immune checkpoint inhibitors, improving the immune response against tumors.

## 7. Clinical Trials of the Interleukin-8/CXCR1/2 Pathway and Immunotherapy Combinations

The IL-8/CXCR1/2 axis has emerged as a critical target in cancer therapy due to its multifaceted roles in tumor growth, angiogenesis, metastasis, and immune suppression (CXCR1/2 inhibitors, Table 2, Figure 2). BMS-986253 (HuMax-IL8) is a fully human monoclonal antibody designed to neutralize human IL-8. In preclinical studies, IL-8 blockade with BMS-986253 has been shown to reduce the recruitment of polymorphonuclear MDSC, revert mesenchymal transition, and enhance susceptibility of cancer cells to immune mediated lysis [92]. In a phase I clinical trial, Bilusic et al. evaluated the safety, tolerability, pharmacokinetics, and preliminary efficacy of BMS-986253 in patients with metastatic or unresectable solid tumors [27]. The study enrolled 15 patients who received escalating doses of BMS-986253 every two weeks to determine the maximum tolerated dose. The trial included detailed monitoring of adverse events and pharmacokinetic analysis, with secondary aim to assess tumor response. Results indicated that BMS-986253 was generally well-tolerated, with treatment-related adverse events occurring in 33% of patients, primarily grade 1, with no dose-limiting toxicities observed. Preliminary efficacy results showed stable disease in some patients, suggesting potential clinical activity. The investigation underscored the role of IL-8 targeting in solid tumors. The authors emphasized the need for further studies to confirm these findings and to explore combination therapies with other immunotherapeutic agents. Several ongoing studies are investigating the combination of BMS-986253 and other immunotherapies including nivolumab for hormone-sensitive prostate cancer (MAGIC-8, NCT03689699), a phase II trial of nivolumab in combination with BMS-986253 or cabiralizumab in advanced hepatocellular carcinoma (NCT04050462), and an investigational immunotherapy study of BMS-986253 with nivolumab in patients with advanced cancers (NCT03400332).

SX-682 is a small-molecule inhibitor of CXCR1/2 and that has also been found to reduce MDSC recruitment. In preclinical CRC models, as KRAS (Ki-ras2 Kirsten rat sarcoma viral oncogene homolog) mutations were linked to CXCR2 axis activation and promotion of an immunosuppressive TME, SX-682 was shown to abrogate MDSC trafficking as well as enhance the efficacy of anti-PD1 (nivolumab) therapy in these models [93]. The STOPTRAFFIC-1 trial (NCT04599140) is a phase I/II study evaluating SX-682 in combination with nivolumab for refractory RAS-mutated microsatellite-stable (MSS) metastatic CRC. Patients either received a dose-escalating regimen of SX-682 in combination with nivolumab, or SX-682 alone every four weeks. The primary objectives of the STOPTRAFFIC-1 trial include determining the safety profile of SX-682 alone and in combination with nivolumab, including the maximum tolerated dose (MTD) and recommended phase II dose. Secondary objectives include overall response rate, PFS, OS, and pharmacokinetic profiles. Efficacy results from the ongoing phase I/II trial are pending [93]. Another ongoing study (NCT04477343) is evaluating SX-682 in combination with nivolumab as maintenance therapy for unresectable pancreatic adenocarcinoma (PDAC). This phase I trial involves a dose-escalation design to determine the MTD of SX-682 when combined with nivolumab. Nine of the planned 20 patients have been enrolled, with initial dose-level completed without dose-limiting toxicities and increase dosing regimen ongoing [94].

As another small-molecule antagonist of CXCR2, AZD5069, reduces neutrophil recruitment to inflammatory sites [96]. Multiple human trials have been completed to evaluate AZD5069 as a potential therapeutic agent in inflammatory conditions including asthma, chronic obstructive pulmonary disease, and bronchiectasis [97,98,99]. Although these studies were not associated with improved clinical outcomes, they demonstrated that the therapy was well-tolerated, with infections and nasopharyngitis being the most common observed adverse events. A phase I/II study is evaluating AZD5069 in combination with durvalumab in patients with advanced hepatocellular carcinoma [100]. The study aims to determine the recommended phase II dose and assess the antitumor efficacy of the combination. The first dose cohort has been completed with no dose-limiting toxicities, and the second dose cohort is currently recruiting [100].

Finally, navarixin, a small-molecule antagonist of CXCR1/2, has shown potential benefits in multiple solid tumors by inhibiting CXCR1/2-mediated pathways [95]. Preclinical evidence suggests that CXCR2 inhibition can reduce tumor progression, lineage plasticity, and enhance the efficacy of immune checkpoint inhibitors [101,102]. A phase II study (NCT03473925) evaluated navarixin in combination with pembrolizumab in patients with advanced or metastatic castration-resistant prostate cancer (CRPC), microsatellite-stable colorectal cancer (MSS CRC) or non-small-cell lung cancer (NSCLC). Median PFS ranged from 1.8 to 2.4 months. The study was closed early due to lack of efficacy. Treatment-related adverse events occurred in 67% of patients, with dose-limiting toxicities including grade 4 neutropenia and grade 3 transaminase elevation, hepatitis, and pneumonitis. Given the high rates of treatment related adverse events, perhaps using a small molecule antagonist of the CXC motif chemokine receptor is too broad, allowing for problematic on target but off-tissue specificities. One additional navarixin study is ongoing and we await their results. This phase 2 basket study (NCT05453825) is investigating navarixin monotherapy or in combination with chemotherapy in patients with select advanced solid tumors, including colorectal cancer. The primary endpoints are objective response rate and PFS, with secondary endpoints including OS and disease control rate [103].

The preclinical and early clinical evidence supports IL-8 and CXCR1/2 inhibition as a rational and promising therapeutic strategy, particularly in tumors characterized by a proinflammatory, neutrophil- and MDSC-enriched microenvironment, such as PC. Given its ability to attenuate immunosuppressive myeloid infiltration, reverse epithelial–mesenchymal transition, and enhance susceptibility to immune-mediated killing, IL-8 blockade is especially well-suited as an adjunct to other modalities but may have efficacy as monotherapy, especially for symptomatic control of malignant ascites.

Timing and context of IL-8 pathway inhibition may be critical. Data from animal models and early-phase clinical trials suggest that its maximal benefit may occur when combined with immune checkpoint inhibitors (ICIs), particularly PD-1/PD-L1 blockade. In these settings, IL-8 inhibition appears to reprogram the TME toward immune activation and improve response rates to ICIs, as demonstrated in murine pancreatic and colorectal cancer models [22,90,91] and supported by ongoing trials such as STOPTRAFFIC-1 (NCT04599140) and the MAGIC-8 study (NCT03689699). These findings imply that IL-8 blockade may sensitize tumors that are otherwise refractory to ICIs by reducing myeloid-driven resistance mechanisms. Conversely, the integration of IL-8 pathway inhibition with chemotherapy is less well-defined. Cytotoxic chemotherapy itself can induce immunogenic cell death and transiently deplete immunosuppressive myeloid cells, potentially overlapping with the mechanism of IL-8 inhibition. Whether this results in synergy or redundancy remains an open question, but clinical trials combining navarixin or AZD5069 with chemotherapy are underway (NCT05453825), and their outcomes will inform this strategy.

The rationale for combining IL-8/CXCR1/2 inhibitors with ICIs stems from the ability of the IL-8/CXCR1/2 axis to create an immunosuppressive tumor microenvironment (TME). IL-8 promotes the recruitment of myeloid-derived suppressor cells (MDSCs), neutrophils, and tumor-associated macrophages (TAMs) into the TME. These cells, in turn, suppress T-cell activity, inhibit dendritic cell maturation, and promote angiogenesis, effectively shielding the tumor from immune attack.

ICIs, such as anti-PD-1/PD-L1 and anti-CTLA-4 antibodies, reinvigorate exhausted T cells and unleash their cytotoxic potential. However, the efficacy of ICIs is often limited by the presence of an immunosuppressive TME. By simultaneously targeting the IL-8/CXCR1/2 axis, we aim to disrupt the immunosuppressive barrier and enhance the ability of ICIs to elicit a robust anti-tumor immune response. In essence, IL-8/CXCR1/2 inhibitors can “prime” the TME, making it more susceptible to ICI therapy.

### Future Directions and Challenges

While the combination of IL-8/CXCR1/2 inhibitors with ICIs holds significant promise, several challenges remain. These include identifying predictive biomarkers to select patients most likely to benefit from this combination therapy, optimizing the dosing and scheduling of IL-8/CXCR1/2 inhibitors in combination with ICIs, and managing potential toxicities associated with combination therapy. Future research should focus on addressing these challenges to maximize the therapeutic potential of targeting the IL-8/CXCR1/2 axis in combination with immunotherapy. Further investigation into the impact on the tumor vasculature by limiting angiogenesis may also prove beneficial.

## 8. Conclusions

This review highlights IL-8 as a central mediator of tumor progression, immune suppression, and inflammation in PC, positioning it as a compelling therapeutic target. IL-8 drives key pathological processes such as EMT, neutrophil recruitment, angiogenesis, and fibrosis—contributing to tumor growth and cause complications like malignant ascites formation and bowel obstructions. Preclinical and early clinical data support IL-8 blockade, particularly in combination with immune checkpoint inhibitors, as a promising approach to reprogram the TME and improve clinical outcomes. These insights highlight the need for continued translational and clinical research to validate IL-8-targeted strategies and integrate them into effective treatment paradigms for PC.

## Figures and Tables

**Figure 1 curroncol-32-00496-f001:**
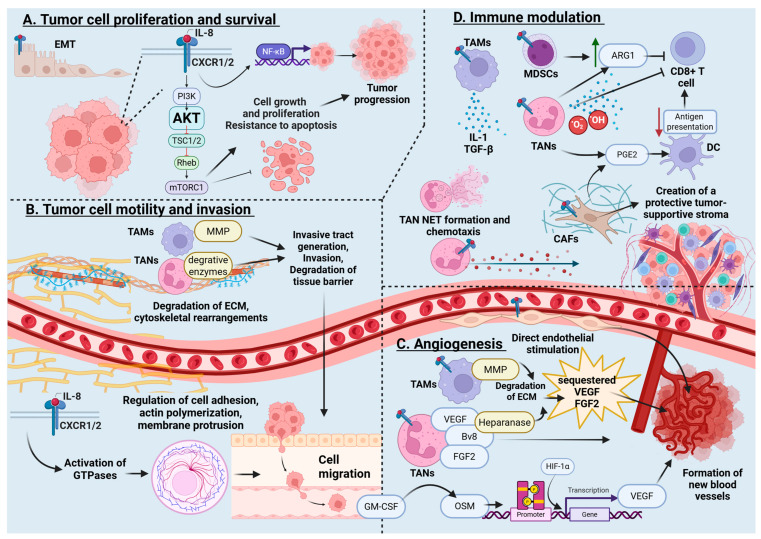
Effects of IL-8 on tumor biology. IL-8 plays a multifaceted and potent role in the pathogenesis and progression of cancer and especially peritoneal carcinomatosis (PC). Through its capacity to induce epithelial–mesenchymal transition, recruit immunosuppressive cells, promote angiogenesis, remodel the extracellular matrix, and support tumor cell survival and dissemination, IL-8 acts as a central orchestrator of PC. Created in BioRender. Wagner, P., (2025) https://BioRender.com/wt0hsjd. Note: EMT, epithelial-to-mesenchymal transition; IL-8, interleukin-8; PI3K, Phosphoinositide 3-kinase; AKT, protein kinase B; TSC1/2, Tuberous Sclerosis Complex 1/2; Rheb, Ras homolog enriched in brain; mTORC, mammalian target of rapamycin complex 1; NF-kβ, Nuclear Factor kappa B; TAMs. Tumor-associated macrophages; MDSCs, myeloid-derived suppressor cells; TANs, tumor-associated neutrophils; ARG1, arginase 1; PGE2, prostaglandin E2; CAFs, cancer-associated fibroblasts; NETs, neutrophil extracellular traps; MMP, matrix metalloproteinase; ECM, extracellular matrix; GM-CSF, Granulocyte-Macrophage Colony-Stimulating Factor; OSM, oncostatin M; VEGF, Vascular Endothelial Growth Factor; FGF2, fibroblast growth factor 2; Bv8, prokineticin 2; IL-1β, interleukin-1 beta; TGF-β, Transforming Growth Factor beta; HIF-1α, hypoxia inducible factor-1 alpha.

**Figure 2 curroncol-32-00496-f002:**
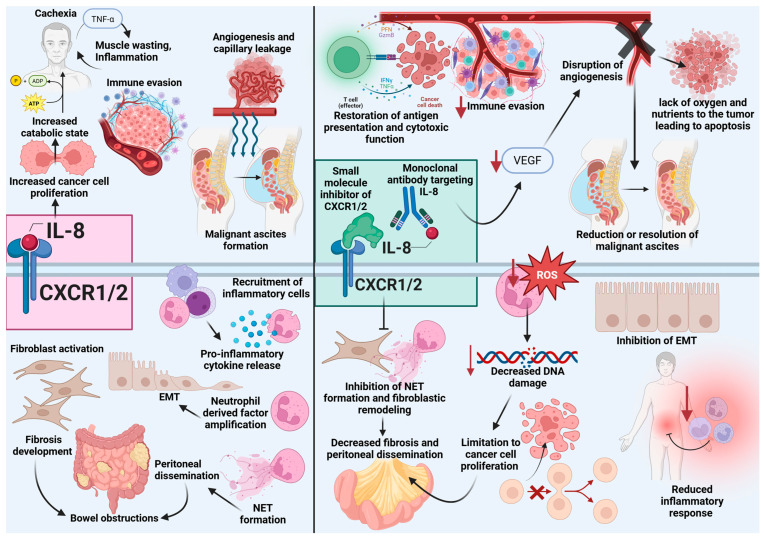
Effects of IL-8 stimulation versus IL-8 inhibition in peritoneal carcinomatosis. On the left, IL-8 signaling through CXCR1/2 promotes tumor progression by enhancing cancer cell proliferation, immune evasion, angiogenesis, fibroblastic activation and subsequent fibrosis, epithelial-to-mesenchymal transition (EMT), neutrophil recruitment and NET formation, peritoneal dissemination, malignant ascites formation, and bowel obstructions. These processes are driven by proinflammatory cytokine release and proliferative transcription mediators, contributing to a catabolic and immunosuppressive state. On the right, inhibition of IL-8/CXCR1/2 signaling- via monoclonal antibodies or small molecule inhibitors- counteracts these effects by disrupting inflammatory feedback loops sustaining the TME, reduce neutrophil and MDSC recruitment, restoring antigen presentation, reducing immune evasion, inhibiting angiogenesis, resolving ascites, limiting ROS-mediated DNA damage, suppressing EMT and NET formation- limiting peritoneal dissemination, and decrease fibrosis. These changes result in reduced proliferation, improved immune responses, and attenuation of inflammation and tumor burden. Created in BioRender. Wagner, P., (2025) https://BioRender.com/vfqbh5f. Note: IL-8, interleukin-8; NET, neutrophil extracellular trap; EMT, epithelial-to-mesenchymal transition; DNA, deoxyribonucleic acid; TNFα, tumor necrosis factor alpha.

**Table 1 curroncol-32-00496-t001:** Various tumor-promoting functions derived from IL-8 axis stimulation and their associated mechanism.

Tumor-Promoting Function	Pathway	Mechanism	Reference
**Proliferation and survival**	-AKT	Promotes tumor cell proliferation and resistance to apoptosis.	[45,46,47,48,49,50]
	-NF-κB	Tumor growth and neutrophil infiltration.	
	-ERRα activation	Tumor cell proliferation and migration.	
**Motility and invasion**	-Small GTPases, cytoskeletal dynamics	IL-8 enhances tumor cell motility by regulating adhesion, actin polymerization, and cytoskeletal rearrangements. Neutrophils degrade tissue barriers, aiding invasion.	[23,51]
	-Activation of proteases	Degradation of tissue barriers.	
**Angiogenesis**	-VEGF, Bv8, MMP-9, JAK-STAT, FGF2	IL-8 directly stimulates endothelial cells for vessel formation. Neutrophils release VEGF, Bv8, and MMP-9, degrading ECM to promote vascularization. FGF2 and OSM further enhance angiogenesis.	[23,50,52,53]
**Epithelial–mesenchymal transition (EMT)**	-NF-κB, HMGB1/TLR4/NF-κB	IL-8 drives EMT by downregulating epithelial markers and upregulating mesenchymal markers. Additional factors like TGF-β, IL-17, and NE amplify EMT. NET formation reawakens dormant tumor cells.	[53,54]
**Microenvironment modulation**	-PI3K/AKT, JAK-STAT3, PD-L1, ARG1, ROS	IL-8 recruits and activates neutrophils, MDSCs, and TAMs, suppressing adaptive immunity and fostering an immunosuppressive TME. PD-L1 expression and ROS production further suppress T-cell function.	[23,25,44,53]

Note: AKT: Ak strain transforming pathway, NF-κB: Nuclear factor kappa-light-chain-enhancer of activated B cells, ERRα: estrogen-related receptor alpha, VEGF: vascular endothelial growth factor, MMP-9: matrix metalloproteinase-9, FGF2: fibroblast growth factor-2, TLR4: toll-like receptor-4, EMT: epithelial-to-mesenchymal transition, NET: neutrophil extracellular traps, PD-L1: programmed death-ligand 1, ARG1: arginase-1, ROS: reactive oxygen species.

**Table 2 curroncol-32-00496-t002:** Review of IL-8 blockade agents and the mechanism of action of monoclonal antibodies and small-molecule inhibitors that impede the IL-8 axis. The completed and ongoing studies are included, as well as the reported adverse event rates for each trial.

Drug	Type	Mechanism of Action	Preclinical Evidence	Completed/Ongoing Studies	Details	Adverse Events	Reference
*BMS-986253*	MAb	Inhibits IL-8, reducing tumor progression and immune evasion	Reduced mesenchymal features in tumor cells	NCT03689699, NCT04050462, NCT03400332	Phase I: 15 patients, 33% adverse events, no dose-limiting toxicities	73% stable disease, median duration 24 weeks	[27]
*SX-682*	SMI	Inhibits CXCR1/2, reducing MDSC recruitment and enhancing anti-PD-L1 efficacy	Abrogated MDSC trafficking, enhanced anti-PD-L1 efficacy	NCT04599140, NCT04477343	STOPTRAFFIC-1: ongoing, no dose-limiting toxicities	None Reported	[93,94]
*Navarixin*	SMI	Inhibits CXCR2, reducing tumor progression and enhancing immune checkpoint efficacy	Potential benefit in multiple solid tumors	NCT03473925, NCT05453825	Phase II: 105 patients, 5% partial response in CRPC, 2.5% in MSS CRC	67% treatment-related adverse events, manageable safety profile	[95]

Note: COPD: Chronic obstructive pulmonary disease, Mab: Monoclonal Antibody, SMI: Small-molecule inhibitor, MDSC: myeloid-derived suppressor cell, PD-L1: programmed death-ligand 1, CRPC: castration-resistant prostate cancer, mCRC: mucinous colorectal cancer, MSS CRC: microsatellite stable colorectal cancer.

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
