# Peer review of "The Interleukin-8-CXCR1/2 Axis as a Therapeutic Target in Peritoneal Carcinomatosis"

_curroncol, 2025, doi:10.3390/curroncol32090496_

Round 1
Reviewer 1 Report
Comments and Suggestions for Authors
This review provides a comprehensive and insightful overview of the IL-8 axis and its involvement in various cancer types. It effectively synthesizes current knowledge regarding IL-8 and its receptors, highlighting their roles in inflammatory processes and cancer-related conditions. The inclusion of clinical trials investigating IL-8 inhibitors and receptor-targeted therapies across different malignancies adds valuable translational relevance. The manuscript is clearly written and engaging. I have a few minor suggestions for improvement.
Title of the manuscript – Although the paper is titled The IL-8 as a Therapeutic Target in Peritoneal Carcinomatosis, the majority of its content focuses on a general overview of the IL-8 axis and targeted therapies across various cancer types. The specific discussion on peritoneal carcinomatosis is limited to a single subsection, and while the involvement of IL-8 in this malignancy is mentioned, it does not constitute the central theme of the review. Therefore, the current title may not fully reflect the scope and emphasis of the manuscript. It might be worth considering a more general title that better captures the broader focus of the review, rather than one specifically centered on peritoneal carcinomatosis.
Figures – The figures are carefully prepared and include a substantial amount of graphical content. However, in my opinion, this density of illustrations and text makes them difficult to read. Simplifying the layout into more schematic representations or dividing the figures into smaller, more focused components could potentially improve clarity and enhance the overall readability.
Author Response
Comment 1: Title of the manuscript – Although the paper is titled The IL-8 as a Therapeutic Target in Peritoneal Carcinomatosis, the majority of its content focuses on a general overview of the IL-8 axis and targeted therapies across various cancer types. The specific discussion on peritoneal carcinomatosis is limited to a single subsection, and while the involvement of IL-8 in this malignancy is mentioned, it does not constitute the central theme of the review. Therefore, the current title may not fully reflect the scope and emphasis of the manuscript. It might be worth considering a more general title that better captures the broader focus of the review, rather than one specifically centered on peritoneal carcinomatosis.
Response 1: We appreciate the reviewer's comment regarding the title. We agree that the current title might be too restrictive, especially considering our broad overview of the IL-8 axis in various cancers. Given the restricted literature on IL-8’s role in peritoneal carcinomatosis we felt it was necessary to incorporate information from various cancers to provide more context for rational as a therapeutic target in peritoneal carcinomatosis. We propose the new title “The Interleukin-8-CXCR1/2 Axis as a Therapeutic Target in Peritoneal Carcinomatosis” to broaden the topic of the IL-8/CXCR1/2 Axis with an emphasis on a potential target in peritoneal carcinomatosis.
Comment 2: Figures – The figures are carefully prepared and include a substantial amount of graphical content. However, in my opinion, this density of illustrations and text makes them difficult to read. Simplifying the layout into more schematic representations or dividing the figures into smaller, more focused components could potentially improve clarity and enhance the overall readability.
Response 2: We thank the reviewer for pointing out the density of information in the figures, particularly Figure 1 and Figure 2. We recognize this may impact readability and agree that a more streamlined presentation would improve clarity. In the revised manuscript, we simplified the schematics in both figures, reducing the number of arrows and text labels directly on the illustrations. Our goal is to improve the visual clarity and make the figures more accessible.
Reviewer 2 Report
Comments and Suggestions for Authors
Authors of the manuscript "The Interleukin-8 Axis as a Therapeutic Target in Peritoneal Carcinomatosis" describe the functional role of IL-8 cytokine and the receptors in neutrophils in Peritoneal Carcinomatosis, a metastatic tumor originating from different organ sites. They have attributed various tumor phenotypes; cell growth, angiogenesis, and metastasis to IL-8. It is known that cytokine IL-8 is transcribed by NFkB which is activated by growth factors such as TNFα. Number of other cytokines including IL-6, and IL-1β are transcribed by NFkB. Authors themselves have published an article on the inhibition of IL-6 in tumor cell growth suppression.
Like NFkB, various transcription factors, and signaling pathway molecules play a role in tumor cell growth, vascularization and metastasis. Thus, attributing all the phenotypes to IL-8 is an oversimplification and needs careful attention. IL-8 may be associated, but it is not a causative agent to suggest inhibition of this molecule will lead to suppression of metastasis.
As the authors have shown, there are only a few pre-clinical and clinical investigations of IL-8 in tumor cell growth suppression. Authors have included its receptors on neutrophils, CXCR1/2 to indicate inhibition of IL-8 axis. This is a simplistic view and the title, and the article needs to reflect inhibition of CXCR1/2 receptors as well as the adjuvant role played by these inhibitors in immunotherapy, the main cancer therapeutic intervention in different human cancers including metastatic tumors.
It is surprising that the authors have included COPD trial of AZD5069 in Table 2 of this article on cancer metastasis to indicate anti-chemotaxis, and anti-inflammatory role of this molecule. Inclusion of anti-inflammatory role of this molecule in cancer will be helpful.
Author Response
Comment 1: Authors of the manuscript "The Interleukin-8 Axis as a Therapeutic Target in Peritoneal Carcinomatosis" describe the functional role of IL-8 cytokine and the receptors in neutrophils in Peritoneal Carcinomatosis, a metastatic tumor originating from different organ sites. They have attributed various tumor phenotypes; cell growth, angiogenesis, and metastasis to IL-8. It is known that cytokine IL-8 is transcribed by NFkB which is activated by growth factors such as TNFα. Number of other cytokines including IL-6, and IL-1β are transcribed by NFkB. Authors themselves have published an article on the inhibition of IL-6 in tumor cell growth suppression.
Like NFkB, various transcription factors, and signaling pathway molecules play a role in tumor cell growth, vascularization and metastasis. Thus, attributing all the phenotypes to IL-8 is an oversimplification and needs careful attention. IL-8 may be associated, but it is not a causative agent to suggest inhibition of this molecule will lead to suppression of metastasis.
Response 1: We acknowledge the reviewer's valid concern that attributing all tumor phenotypes to IL-8 could be seen as an oversimplification. We agree that tumor development is a complex, multifactorial process. In the revised manuscript, we were more precise in our language. We rephrased statements to avoid implying that IL-8 is the sole causative agent. We used language that emphasizes its significant contribution and interaction with other pathways (e.g., NF-kB, TNFα) in driving these processes. (change has been done in section 2 and 3 of paper)
Comment 2: As the authors have shown, there are only a few pre-clinical and clinical investigations of IL-8 in tumor cell growth suppression. Authors have included its receptors on neutrophils, CXCR1/2 to indicate inhibition of IL-8 axis. This is a simplistic view and the title, and the article needs to reflect inhibition of CXCR1/2 receptors as well as the adjuvant role played by these inhibitors in immunotherapy, the main cancer therapeutic intervention in different human cancers including metastatic tumors.
Response 2: The reviewer makes an excellent point regarding the therapeutic focus extending to both IL-8 and its receptors, CXCR1/2. We reflected this more explicitly in the title. We propose: " The Interleukin-8-CXCR1/2 Axis as a Therapeutic Target in Peritoneal Carcinomatosis". Furthermore, we agree that the potential adjuvant role of IL-8/CXCR1/2 inhibitors in immunotherapy warrants greater emphasis. The text in Section 7 (Clinical Trials of IL-8 Pathway Inhibition) was revised to: 1) Explicitly discuss the rationale for combining these inhibitors with ICIs. 2) Highlight clinical data (where available) suggesting synergistic effects.
Comment 3: It is surprising that the authors have included COPD trial of AZD5069 in Table 2 of this article on cancer metastasis to indicate anti-chemotaxis, and anti-inflammatory role of this molecule. Inclusion of anti-inflammatory role of this molecule in cancer will be helpful.
Response 3: We understand the reviewer's concern regarding the inclusion of the COPD trial of AZD5069 in Table 2. While our intention was to highlight the well-established anti-chemotactic properties of AZD5069 in a clinical setting, we acknowledge that its inclusion in a table focused on cancer trials may be misleading. We will remove the AZD5069 COPD trial data from Table 2 to avoid confusion and maintain a clear focus on cancer-related studies. The relevant information about anti-chemotaxis will be mentioned in the main text.
Round 2
Reviewer 1 Report
Comments and Suggestions for Authors
I confirm my acceptance of the work. I appreciate the corrections made.
Author Response
Comment 1: I confirm my acceptance of the work. I appreciate the corrections made.
Response 1: Thank you for your kind feedback and for confirming your acceptance of our work, We greatly appreciate the time and effort you devoted to reviewing our manuscript and your recognition of the corrections made.
Reviewer 2 Report
Comments and Suggestions for Authors
The revised manuscript has addressed the issues raised. Authors could add CXCR1/CXCR2 receptor in the simple summary as well.
In lines 18/19: IL-8 attracts immune cells like neutrophils that....create an environment through its receptor CXCR1/CXCR2 signaling.....avoid immune attack.
lines 24/25: Our review concludes that targeting IL-8 and its receptor CXCR1/CXCR2 is a promising.....
Author Response
Comment 1: The revised manuscript has addressed the issues raised. Authors could add CXCR1/CXCR2 receptor in the simple summary as well.
In lines 18/19: IL-8 attracts immune cells like neutrophils that....create an environment through its receptor CXCR1/CXCR2 signaling.....avoid immune attack.
lines 24/25: Our review concludes that targeting IL-8 and its receptor CXCR1/CXCR2 is a promising.....
Response 1: The reviewer makes an excellent point of additional points within the simple summary text to further highlight the CXCR1/2 receptor target. We have incorporated the receptor into both sentences within the simple summary as suggested. We thank you for your kind feedback and greatly appreciate your time and effort devoted to reviewing our manuscript.